# Radiation-Induced Nucleation and Growth of CaSi_2_ Crystals, Both Directly during the Epitaxial CaF_2_ Growth and after the CaF_2_ Film Formation

**DOI:** 10.3390/nano12091407

**Published:** 2022-04-20

**Authors:** Anatoly V. Dvurechenskii, Aleksey V. Kacyuba, Gennadiy N. Kamaev, Vladimir A. Volodin, Zhanna V. Smagina

**Affiliations:** 1Rzhanov Institute of Semiconductor Physics, Siberian Branch of Russian Academy of Sciences, 630090 Novosibirsk, Russia; kacyuba@isp.nsc.ru (A.V.K.); kamaev@isp.nsc.ru (G.N.K.); volodin@isp.nsc.ru (V.A.V.); smagina@isp.nsc.ru (Z.V.S.); 2Physics Department, Novosibirsk State University, 630090 Novosibirsk, Russia

**Keywords:** crystal structure, nanostructures, electron irradiation, molecular-beam epitaxy, calcium compounds, semiconducting silicon

## Abstract

The radiation-induced phenomena of CaSi_2_ crystal growth were investigated, both directly during the epitaxial CaF_2_ growth on Si (111) and film irradiation with fast electrons on Si (111) after its formation, while maintaining the specified film thickness, substrate temperature and radiation dose. Irradiation in the process of the epitaxial CaF_2_ film growth leads to the formation of CaSi_2_ nanowhiskers with an average size of 5 µm oriented along the direction <110>. The electron irradiation of the formed film, under similar conditions, leads to the homogeneous nucleation of CaSi_2_ crystals and their proliferation as inclusions in the CaF_2_ film. It is shown that both approaches lead to the formation of CaSi_2_ crystals of the 3R polymorph in the irradiated region of a 10 nm thick CaF_2_ layer.

## 1. Introduction

Investigations of the growth and properties of calcium silicides are currently being carried out by some groups of researchers, and calcium silicide films have been obtained on Si (111) and Si (001) surfaces with various phase compositions. It has been found that, when Ca and Si are co-deposited on a hot substrate with an increasing substrate temperature, there is a tendency to form a phase with the highest silicon content: Ca_2_Si-CaSi-CaSi_2_ [1,2,3]. It has been established that Ca_2_Si films are a narrow-gap semiconductor material, while CaSi and CaSi_2_ films are semimetals. Ca_2_Si showed the best thermoelectric properties [4,5,6,7]. Approaches are being developed to obtain materials with different functional properties: translucent and conductive CaSi_2_ films for silicon electronics and optoelectronics [7,8,9,10]. Recently we reported on the studies of the 20 keV electron-beam irradiation (current density 50 µA/m^2^) effects on the epitaxial CaF_2_ film growth on a Si surface. It was found that the area exposed to the electron beam during the epitaxial growth of the CaF_2_ film is the synthesized CaSi_2_ layer. Radiation-stimulated CaSi_2_ growth allows local CaSi_2_ films to be prepared on a Si wafer and the sizes of CaSi_2_ regions to by limited purely by size of the focused electron beam and has the possibility to control the CaSi_2_ film thickness with the accuracy of one monolayer, by varying both the radiation dose and the initial Si layer thickness as well as the sacrificial CaF_2_ layer [8].

The increased interest in CaSi_2_ is due to the possibility of obtaining high-quality epitaxial CaSi_2_ films on silicon substrates, which makes it possible to manufacture devices based on this material with standard silicon planar technology [11,12]. In addition to the possibility of epitaxial growth on Si, CaSi_2_ has a unique layered crystal structure consisting of a hexagonal Si bilayer and a trigonal Ca monolayer [13,14,15], which makes it possible to consider this material as a prototype for obtaining 2D structures based on Si. Thus, it was found that silicon layers intercalated in CaSi_2_ exhibit electronic properties similar to those of graphene [16].

Usually, to obtain epitaxial CaSi_2_ films on Si, atomic Ca is applied, and the substrate is heated, which leads to the Ca interaction with Si [17,18]. When studying CaF_2_ film growth on Si (111) by molecular-beam epitaxy (MBE) under the conditions of simultaneous irradiation with an electron beam with the energy of 20 keV and current density of 50 µA/m^2^, we found that the radiation-stimulated CaSi_2_-film growth is observed at the interface of the silicon substrate and the epitaxial CaF_2_ film [19]. The application interest in the method of radiation-stimulated CaSi_2_-film growth is related to the possibility of forming a local CaSi_2_ film area on the plate, the size of which is limited only by the focused electron beam diameter. There is also interest in controlling the thickness up to the monolayer by changing both the radiation dose and the thickness of the initial Si layer and the sacrificial CaF_2_ layer.

Several related CaSi_2_ phases are known: a single-layer phase (the so-called EuGe2 structure, spatial group P3m1), a two-layer phase (spatial group P63mc, which exists only under pressure and has a superconductivity at temperatures below 14 K), and polymorph structures with a three-layer and six-layer translational period of silicon substructures in the unit cell 3R and 6R, respectively (spatial group R3m for both compounds) [20,21]. In [22], calcium silicide films were obtained in an ultrahigh vacuum installation by the co-deposition of Ca and Si (MBE method). The authors found that calcium silicide films grown at temperatures of 300–330 °C using the MBE method on Si (111) substrates are a homogeneous mixture of amorphous and nanocrystalline CaSi_2_. The heteroepitaxial CaSi_2_ growth is observed on the Si (111) substrate at 500 °C during the deposition of calcium atoms (reactive epitaxy method), which leads to the formation of 6R-CaSi_2_ (001)/Si (111).

The aim of this work is to establish the processes of nucleation and growth of CaSi_2_ crystals under the influence of an electron beam on CaF_2_, both directly during the epitaxial CaF_2_ growth and after the CaF_2_ film formation, while maintaining the same deposited film thickness, radiation dose, and substrate temperature.

## 2. Materials and Methods

The experiments were carried out in the «Katun-100» molecular-beam epitaxy (MBE) [23] unit, under ultrahigh vacuum conditions, equipped with a CaF_2_ effusion source with a graphite crucible. Si (111) plates of 100 mm in diameter were used as substrates. For the purpose of preliminary cleaning, the substrates were annealed in the vacuum chamber for 6 h at a temperature of 400 °C. Further, at a temperature of 720 °C and a weak Si flow, the protective oxide was peeled off until a superstructure (7 × 7) appeared, after which a 100 nm thick buffer Si layer was grown at a temperature of 550 °C.

The epitaxial CaF_2_ film growth was carried out at the deposition rate of 0.3 Å/s at a substrate temperature of 550 °C. For the whole time, the selected section of the CaF_2_ film was irradiated by a RHEED electron beam in the crystallographic direction <110> at an acceleration voltage of 20 keV and a current density of 50 µA/m^2^. The beam incidence angle was 4°. Following this, the substrate was rotated by 60°, and the already-formed 10 nm thick CaF_2_ film was exposed to the electron irradiation for 5 min, which corresponds to the growth time of the 10 nm CaF_2_ film.

The crystal structure of the deposited layers was studied using the RHEED method (in situ) and Raman scattering (RAMAN). The Raman spectra were recorded at room temperature on a T64000 spectrometer manufactured by Horiba Jobin Yvon, and a solid-state laser with a wavelength of 514.5 nm was used. The surface morphology was studied using an atomic force microscopy (AFM) Solver PRO and a scanning electron microscope (SEM) Hitachi SU8220, as well as spectroscopy.

## 3. Results

An example of the optical images of the studied samples is shown in Figure 1. The characteristic light bands L1 and L2 rotated relative to each other at the angle of 60° are observed in the places of exposure to the electron beam. Point L0 is the place where the Raman spectrum was recorded in non-irradiated regions. Band L1 corresponds to the irradiation region during the CaF_2_ film growth, and band L2 corresponds to the electron beam area after the film growth.

The Raman spectra recorded outside of the irradiation region (L0), in the electron beam incidence area during the film growth (L1) (corresponds to the L1 line in Figure 1) and in the electron beam incidence area after the film growth (L2) (corresponds to the L2 line in Figure 1) are shown in Figure 2.

The spectrum from the initial Si substrate completely coincides with the spectrum from the CaF_2_ film recorded in the non-irradiated region (curve L0). The spectrum shows a very intense peak at 520.5 cm^−1^ from the scattering by long-wavelength optical phonons in silicon and the features from 200 to 450 cm^−1^ associated with the two-phonon scattering by acoustic phonons in silicon. The most intense point on the graph is the line at 305 cm^−1^—the 2TA peak. The electron irradiation of the L1 and L2 regions with the same dose at the same temperature led to the appearance of characteristic peaks in the Raman scattering in the regions of 413 cm^−1^, 386 cm^−1^, and 344 cm^−1^, respectively. These peaks are characteristic of the CaSi_2_ polymorph 3R [24]. It is known that the 6R CaSi_2_ polymorph is characterized by the appearance of additional peaks in the 150–250 cm^−1^ spectral region, with a characteristic peak of 203 cm^−1^, due to a decrease in the crystallographic symmetry of Ca [25].

The peak positions coincide for CaSi_2_ films obtained both by deposition of CaFa_2_ on Si (111) under conditions of simultaneous irradiation with fast electrons and by the irradiation with fast electrons of the epitaxial CaF_2_ film after its formation. At the same time, for the Raman spectra of the L1 and L2 regions, different peak amplitudes are observed in the 413 cm^−1^ and 386 cm^−1^ regions. Since the irradiated regions were formed by rotating the substrate by 60° (Figure 1), the difference in the peak amplitudes on the spectra can be related both to the dependence of the intensity of Raman spectra on the polarization directions of the incident and scattered light relative to the crystallographic axes and to the morphological features of the CaSi_2_ crystal film formation associated with a different method of production. To identify the angular dependence of the intensity and the correspondence of the tensors of symmetry groups with the Raman peaks of the CaSi_2_ polymorph 3R, an additional study was carried out using the Raman method at different incident and scattered light polarizations.

For CaSi_2_, the Raman polarization dependence is described by tensors for the spatial groups of *E_g_*(Si), *A*_1*g*_(Si), and *A*_2*g*_(Si_2_), respectively [25].

The peak intensity of the Raman spectra is determined by the ratio: $$I\sim {\left|\sum_{\rho ,\sigma =x,y,z}e_{i}R_{\rho \sigma }e_{s}\right|}^{2},$$
where $e_{i}\left(e_{s}\right)$—incident (scattered) light polarization and $R_{\rho \sigma }$—Raman scattering. The tensors for the groups of symmetry *A*_1*g*_ and *E_g_*(1) are as follows:$$A_{1g}=\left(\begin{array}{ccc} a & 0 & 0\\ 0 & a & 0\\ 0 & 0 & b \end{array}\right)\;,\;E_{g}\left(1\right)=(\begin{array}{ccc} c & 0 & 0\\ 0 & -c & d\\ 0 & d & 0 \end{array})\;$$

Therefore, if incident and scattered light polarizations are parallel to each other, then the mode *E_g_*(1) intensity depends on the direction of the polarization relative to the crystallographic axes and, accordingly, depends on the rotation angle of the sample around the normal. The mode *A*_1*g*_ is allowed in all parallel polarizations and prohibited in all crossed polarizations.

In Figure 3, the Raman spectra are recorded at the irradiation site for the L1 region, with different incident and scattered light polarizations, where the X’ and Y’ axes are rotated 45° relative to XY. In polarization measurements of Raman scattering, the so-called Porto notation is used. In our case, backscattering geometry was used, and the wave vector of the incident light was directed along the Z axis (i.e., along the (111) crystallographic direction in the Si substrate). The wave vector of the scattered light was directed along the -*Z* axis. The light wave is transverse, so, the directions of polarization of the incident and scattered light are perpendicular to the Z axis. In Porto XY notation, the first direction is the polarization of the incident wave, and the second direction is the polarization of the scattered wave. In our case, the X axis corresponds to the crystallographic direction (1–10), and the Y axis corresponds to the crystallographic direction (11–2). One can see that the X, Y and Z axes are orthogonal. The results of the 45° rotation are that the X’ axe is a bisector of the angle X0Y, and the Y’ is a are bisector of the angle X0Y. The XY (or X’Y’) polarization geometry is called ‘transverse’ (i.e., the polarization of scattered light is perpendicular to the polarization of the incident light), and the YY (or Y’Y’) polarization geometry is called ‘longitudinal’, (i.e., the polarization of light does not change upon scattering).

In the L1 region, a change in polarization (Figure 3) leads to a change in the intensity of the main peaks from CaSi_2_ in the region of 413 cm^−1^, 386 cm^−1^, and 344 cm^−1^ at all angles of incident and reflected light. Changes in the intensity of the peaks at all positions indicate that the irradiated region has a certain orientation of CaSi_2_ crystals. The Raman spectra were recorded at the L2 irradiation site with different polarizations of incident and scattered light, where the X’ and Y’ axes were also rotated 45° relative to XY.

Unlike the spectra in Figure 3, where the intensity of peaks from CaSi_2_ in the regions of 413 cm^−1^, 386 cm^−1^, and 344 cm^−1^ varies in all polarization geometries, there are two pairs of spectra in Figure 4. The positions of the peaks in the X’Y’ coordinate planes completely coincide. Consequently, irradiation after growth leads to the formation of orientation-unrelated crystallites with the initial substrate. It should be noted that, for the L1 and L2 regions, the characteristic Raman peaks of CaSi_2_ polymorph 6R crystals do not appear in the 150–250 cm^−1^ spectral region. As for the difference between the observed frequency of *E_g_* mode (418 cm^−1^) and the literature data (413 cm^−1^), one can assume that it occurs due to strain. It is known that, due to anharmonism, phonon frequencies depend on strain. Usually, the compressive strain leads to a blueshift in phonon frequencies.

An RHEED study for two approaches to the synthesis of CaSi_2_ in the L1 and L2 regions was carried out. In Figure 5, the RHEED picture of the initial Si (111) surface after removing the protective oxide and the Si buffer layer growth is depicted. Elongated reflections are visible. There is a 7 × 7(1) superstructure characteristic of Si (111). Kikuchi lines, indicating an atomically smooth monocrystalline Si surface, can also be observed.

The RHEED images after 5 min of irradiation are shown in Figure 6, for the (L1) region—Figure 6a and for the L2 region—Figure 6b.

In both cases, elongated reflections are present in the RHEED pictures after 5 min of electron irradiation of the L1 (a) and L2 (b) regions (Figure 6), indicating an epitaxial monocrystalline film. At the same time, there are a number of differences in the RHEED pattern of the sample irradiated after the CaF_2_ (L2) film growth: (1) concentric semicircles appear, indicated by arrow 3 (see inset to Figure 6bL2), and (2) double reflections are indicated by arrow 1 and 2. The presence of concentric semicircles clearly indicates the formation of a polycrystalline phase. The appearance of double reflection in the RHEED picture is due to the fact that the film contains clusters with different crystal lattice constants. Since the already formed CaF_2_ film was exposed to radiation in the L2 region, at the initial moment of time we observed the RHEED pattern from the CaF_2_ film. The dose dependence of the RHEED pattern on the L2 region after 1 min of irradiation (Figure 7a) and after 5 min of irradiation (Figure 7b) is shown in Figure 7.

At low irradiation doses and due to the small amount of the CaSi_2_ phase formed, there are no concentric semicircles and double reflections noted in Figure 7a. At the same time, the Raman data indicate that the CaF_2_ film irradiation with an electron beam at the temperature of 550 °C leads to the formation of CaSi_2_. Therefore, the appearance of double reflection and concentric semicircles in the RHEED image after 5 min of irradiation is associated with the formation of CaSi_2_. The presence of simultaneously elongated reflections and concentric half-rings in the RHEED image indicates that irradiation after CaF_2_ film growth leads to the formation of CaSi_2_ crystallites, which are disoriented relative to each other, which is also confirmed by the Raman spectra with different polarizations of incident and scattered light for the L2 band (Figure 4).

In Figure 8, the RHEED patterns from the region (L1) at the initial moment (20 s) of irradiation and growth (a) and 1 min of irradiation and growth (b) are shown. When irradiated during growth, the RHEED pattern is changed only at the initial moment of time during the transition from the Si substrate to the growing film. In Figure 8a, there is still a superstructure from the Si substrate indicated by arrow 1 and the Kikuchi lines (arrow 2). After 1 min of irradiation, only the Kikuchi lines of Figure 8b are observed, and they disappear during the ongoing irradiation. At the same time, the main reflections remain unchanged throughout the entire growth (5 min), and they are associated with the formation of the CaSi_2_ phase.

Further, there are photographs of the surface obtained using the AFM and SEM methods. Figure 9 shows the initial surface of the epitaxial CaF_2_ film and its profile, which was not irradiated by an electron beam (region L0).

It is shown in Figure 9 that the initial CaF_2_ film has a surface roughness less than 1 nm. There are triangular islands characteristic of CaF_2_ on the surface. Figure 10 shows the SEM-images at the site of electron irradiation (regions L1 and L2) with different magnifications.

It can be seen in Figure 10 that two different methods of producing CaSi_2_ (electron irradiation during growth and electron irradiation after growth) lead to significant differences in the surface morphology. Irradiation during growth (Figure 10a) leads to the formation of separate islands of different shapes. The AFM images and the surface profile for the L1 region are shown in Figure 11.

Two types of islands are observed on the L1 surface: islets of chaotic shape with an average lateral size of 200–300 nm, 4–6 nm high and islets as nanowhiskers with an average length of 0.5 to 1 µm and 6–10 nm high, which have a strict orientation and line up along the direction of {110}, due to the heterogeneous nucleation on the Si (111) substrate and subsequent growth.

Electron irradiation after CaF_2_ growth (region L2) leads to the formation of CaSi_2_ nanowhiskers with an average size of 5 µm, inside which there are separate islands of 60–70 nm in size and 15–20 nm in height. At the same time, it is impossible to distinguish any predominant directions or their connections with the Si (111) substrate. The surface profile was obtained using the AFM method and is shown in Figure 12. When the formed CaF_2_ film is irradiated with an electron beam, a chaotic homogeneous nucleation occurs, followed by the crystallite growth in all directions.

The average surface roughness in the irradiation area for the region L1 is 6–8 nm and for L2—25–30 nm.

## 4. Discussion

It was found that irradiation during the epitaxial CaF_2_-film growth leads to the heterogeneous nucleation and growth of Si-related nanowhiskers CaSi_2_, oriented along the direction <1–10> with average size 5 um.

Irradiation of the formed epitaxial CaF_2_ film with a thickness of 10 nm at a similar electron irradiation dose and substrate temperature leads to random CaSi_2_ nucleation centers in CaF_2_ film with a subsequent crystallite proliferation. Analysis of the RHEED and SEM patterns shows that disordered CaSi_2_ crystallites are present on the surface. This indicates the homogeneous nucleation of CaF_2_. Various methods of CaSi_2_ (L1, L2) synthesis for thin films (10 nm) lead to the formation of the CaSi_2_ polymorph 3R, despite the fact that this phase is metastable compared to the 6R polymorph. The reason for the metastable phase growth CaSi_2_ polymorph 3R is, apparently, that the electron beam irradiation is a strongly non-equilibrium action during irradiation and further relaxation with rather a short time to form the more complicate structure of 6R-CaSi_2_ (stable phase). For thick CaF_2_ film deposition (more 20 nm), electron irradiation leads to the formation of the CaSi_2_ as a stable 6R polymorph [8].

It must be noted that RHEED was problematic to use for analysis of the growth of CaF_2_ on Si, because the area exposed to the electron beam suffers strong modifications, such as in the surface morphology and film chemical composition. There is information indicating that, under the influence of an electron beam, the phenomenon of radiolysis is observed—the dissociation of films into calcium and fluorine [26].

## 5. Conclusions

CaSi_2_ crystal nucleation and growth under fast electron beam irradiation with an energy of 20 keV and a current density of 50 µA/m^2^ were established both during the epitaxial growth of CaF_2_ and after the formation of a 10 µm thick CaF_2_ film, while maintaining the same deposited film thickness, radiation dose, and substrate temperature.

The film deposition (more 20 nm) electron irradiation led to the formation of the CaSi_2_ as a stable 6R polymorph [8].

## Figures and Tables

**Figure 1 nanomaterials-12-01407-f001:**
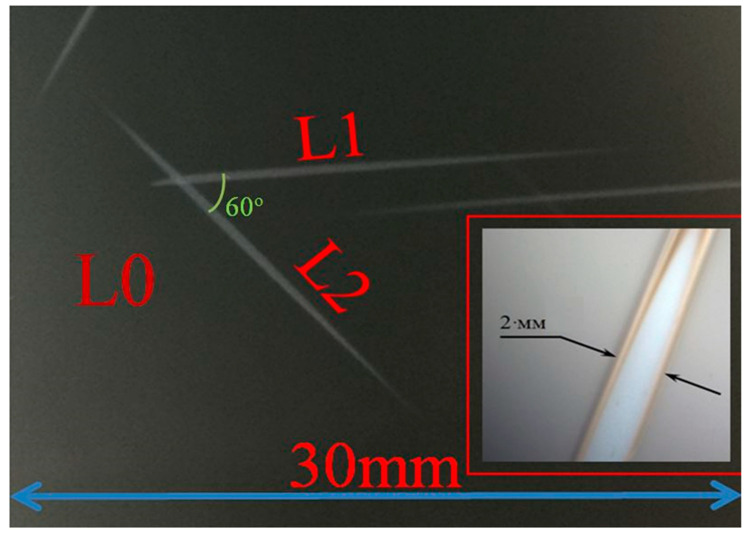
The optical image of trace from the impact of a rapid electron beam on the 10 nm thick CaF_2_ film surface. L1—trace for case electron-beam irradiation directly during CaF_2_ deposition and L2—electron-beam irradiation after the CaF_2_ film formation. In the inset is an enlarged image of the electron-beam track. L0 is no irradiated area of sample with the 10 nm CaF_2_ film thickness.

**Figure 2 nanomaterials-12-01407-f002:**
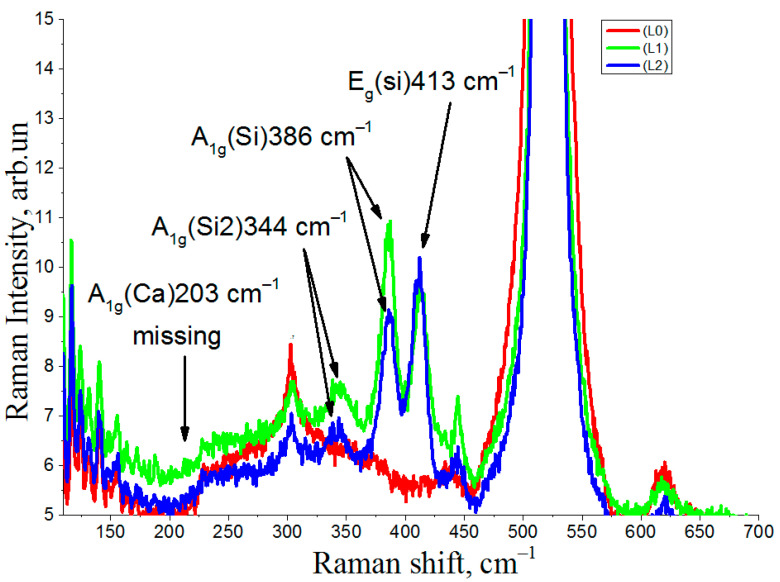
Raman scattering spectra: L0—in the non-irradiated region, L1—irradiation of CaF_2_ during growth, L2—irradiation of CaF_2_ after growth.

**Figure 3 nanomaterials-12-01407-f003:**
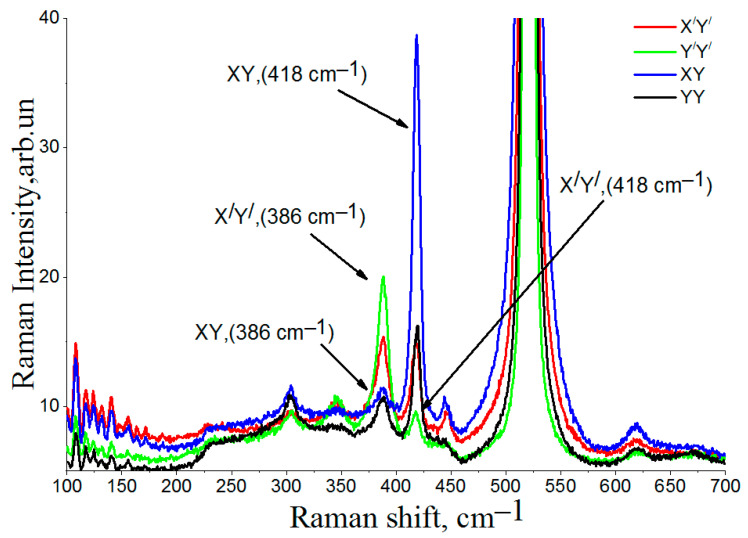
Polarized Raman scattering spectra with different polarizations of incident and scattered light for the L1 region.

**Figure 4 nanomaterials-12-01407-f004:**
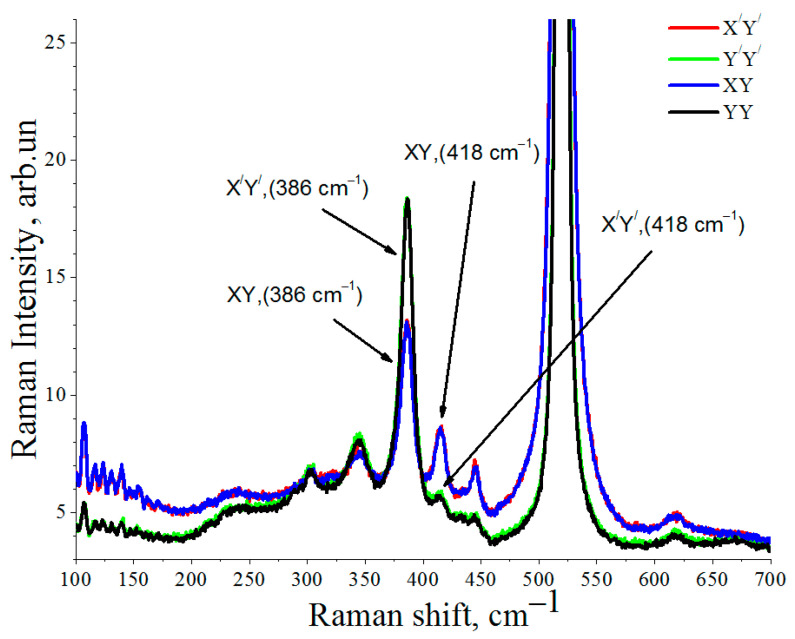
Polarized Raman scattering spectra with different polarizations of incident and scattered light for the L2 region.

**Figure 5 nanomaterials-12-01407-f005:**
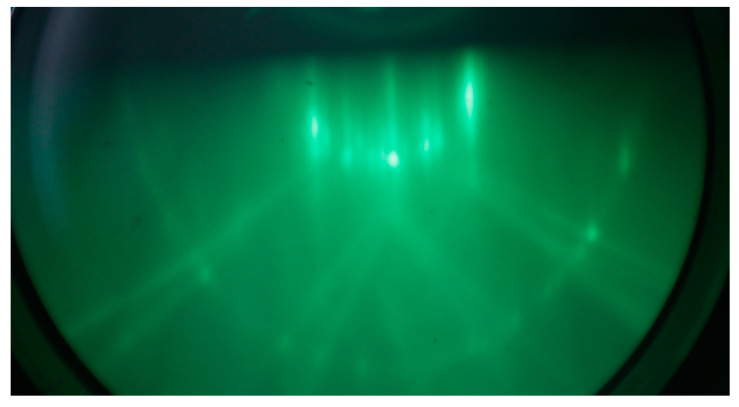
RHEED image of the Si substrate surface after removing a protective oxide and the Si buffer layer growth.

**Figure 6 nanomaterials-12-01407-f006:**
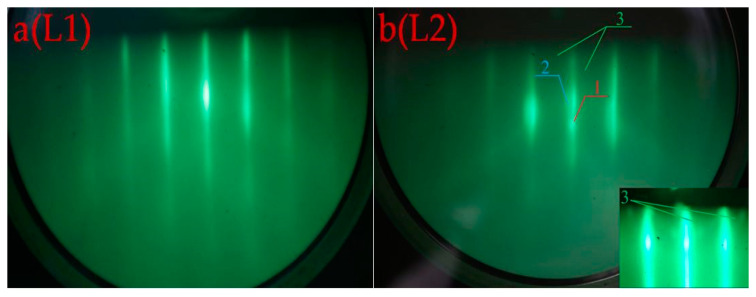
RHEED images after 5 min of irradiating regions L1 (**a**) and L2 (**b**) with electrons.

**Figure 7 nanomaterials-12-01407-f007:**
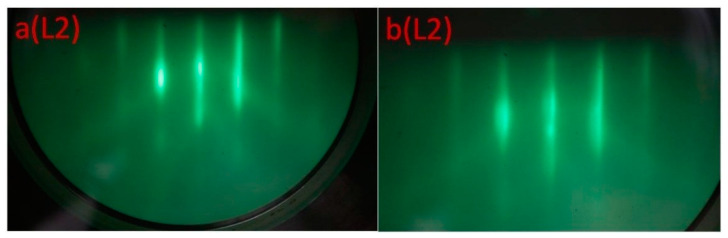
Dose dependences of the RHEED image taken for the L2 region (**a**) after 1 min of irradiation and (**b**) after 5 min of irradiation.

**Figure 8 nanomaterials-12-01407-f008:**
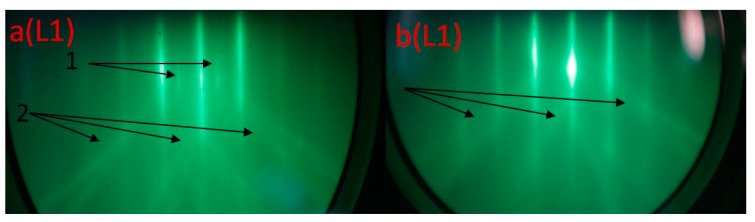
RHEED images taken for region L1: (**a**) after 20 s of irradiation and (**b**) after 1 min of irradiation.

**Figure 9 nanomaterials-12-01407-f009:**
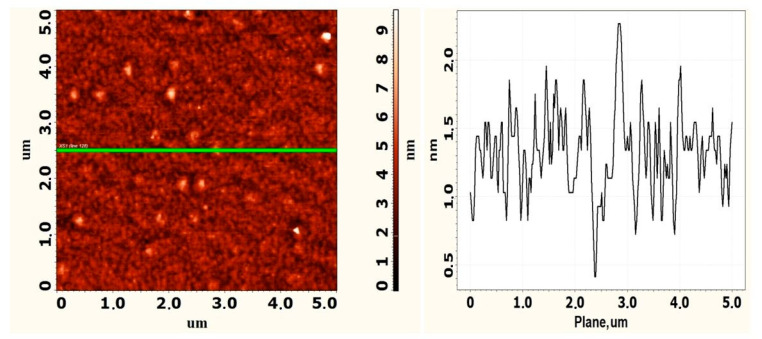
AFM image and profile of the CaF_2_ film surface not exposed to radiation.

**Figure 10 nanomaterials-12-01407-f010:**
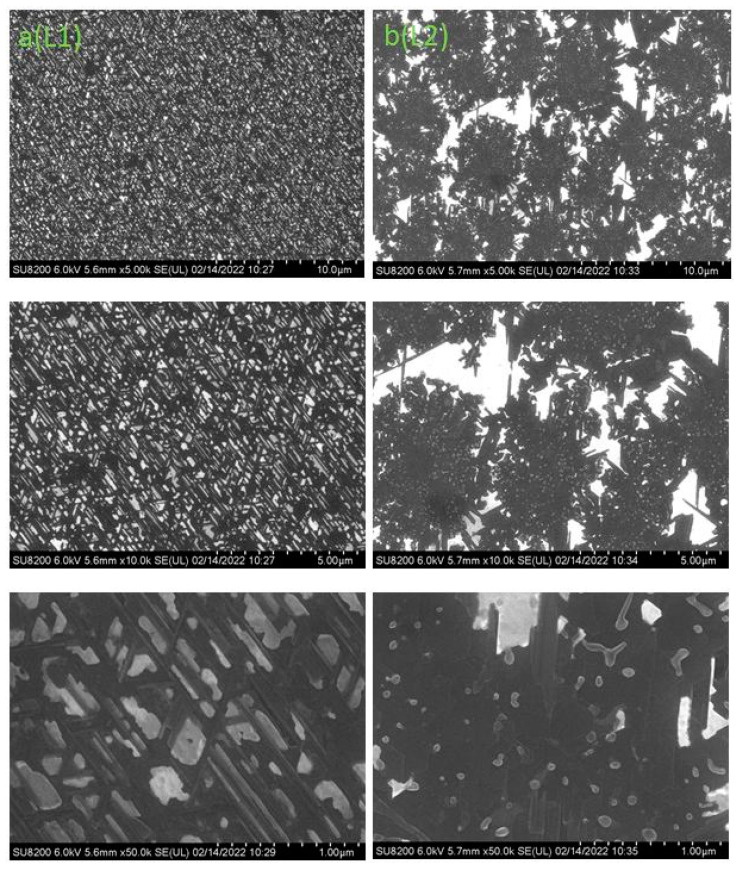
SEM image of the irradiated region: (**a**) during growth (L1)—on the left and (**b**) the irradiated region after the CaF2 growth (L2)—on the right with a characteristic scale of 10 µm, 5 µm, and 1 µm, respectively.

**Figure 11 nanomaterials-12-01407-f011:**
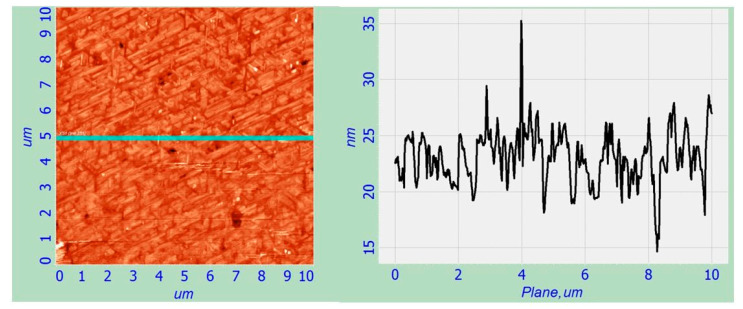
AFM surface image and the surface profile taken in the line for region L1.

**Figure 12 nanomaterials-12-01407-f012:**
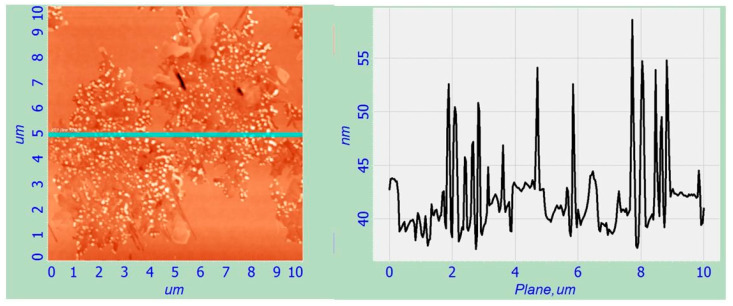
AFM surface image and the surface profile taken in the line for region L2.

## Data Availability

Not applicable.

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
