# Peer review of "Radiation-Induced Nucleation and Growth of CaSi2 Crystals, Both Directly during the Epitaxial CaF2 Growth and after the CaF2 Film Formation"

_nanomaterials, 2022, doi:10.3390/nano12091407_

Round 1
Reviewer 1 Report
The manuscript by A.V. Dvurechenskii et al. presents observations related to silicide formation upon electron beam irradiation of epitaxial CaF2 films on Si(111). Although the manuscript is relatively well written, I have several concerns preventing its acceptance for publication without significant reworking.
As far as I can judge from the Introduction, interest for CaSi2 seems to be marginal, thus more justification of the relevance of this research is suggested. I recommend to extend the Introduction by more up-to-date references to show why this work may be important for the readers of Nanomaterials. In particular, what specific application do the authors see for CaSi2 crystals either directly grown on Si or formed by irradiation of a CaF2 layer?
Section 2: Please provide more details about the MBE apparatus, RHEED, electron gun or at least indicate a reference where these details are available. What was the spot size of the electron gun? Was the spot scanned over the surface? From the text it may appear that the RHEED electron source was used for irradiation; please confirm.
Results, Figs 3-4: even if the qualitative conclusion that irradiation during deposition results in directed crystal growth while post-deposition irradiation leads to formation of randomly oriented crystallites is supported by the data, the presentation is very confusing. Especially, the meaning of the X,Y, X' and Y' axes, their relation to laboratory and Si substrate-fixed coordinate systems is unclear. The meaning of the "2" or "3" numbers in the peak assignments is also unclear. If the Eg mode is at 413 cm-1, why peaks appear at 418 cm-1? I would recommend to reformulate this part (maybe along with a more mathematically correct description of the geometry- and polarization dependence of the Raman intensities), and if it is too complicated or lengthy, to present it as supplementary information.
Figures 6,7: maybe because of the lower quality images provided for review, I'm unable to see the semi-circles. Moreover, the red circles in Fig. 6b make it really difficult to compare the two patterns and to recognize the splitting of the reflections. Please try to resolve the situation!
Although the material formed during electron irradiation of CaF2 was investigated by several different techniques, phase identification is based solely on the Raman spectra. How formation of mixed compounds (like CaSi(2-x)Fx) or silicides with different stoichiometry is ruled out? What happens with the material upon air exposure (as far as I understand, Raman measurements were performed ex situ).
For me a Discussion section is missing. This section would offer opportunity to explain the results, analyze the effect of the different conditions on the film structure and to relate the observations to previous investigations. For example, a relevant question would be the effect of temperature or the electron energy on the silicide formation upon electron beam irradiation. The text currently in Lines 252-261 belongs to this Discussion section. As a side note, do the results presented here imply that RHEED cannot be used for analysis of the growth of CaF2 on Si because of the radiation-induced silicide formation?
Apart from these major points, there are a few minor issues which still require attention:
Lines 36 and 69: the irradiation current density was 50 uA/m2 or 50 uA/cm2?
Lines 80, 86 (Figure 1): with what technique was the image taken? If I understand correctly, the lines represent the irradiated areas, while L0 is the dark uniform area around and between the lines. If it is correct, the white-green dot is confusing.
Figure 2: the notation for the (missing) A1g(Ca) peak is misleading; maybe an arrow to its expected position would be helpful.
Figure 5: was the diffraction pattern recorded after oxide removal or buffer layer growth?
Line 230: please resolve the abbreviation NW.
In general, the English usage throughout the paper is appropriate. However, some revision is still needed. For example:
Line 76: the word "structures" refers to what? Line 80: "image of the studied samples" is the correct wording. Line 90: "recorded outside of the" would be correct. Line 101: "the line at 305 cm-1" would be correct. Lines 102, 111: repeating the same information is unnecessary. Line 158, 167, etc.: please write "reflections" instead of "reflexes". Line 242: "a" is not needed. Lines 257-259: please reconsider the sentence.
Author Response
Point 1: As far as I can judge from the Introduction, interest for CaSi2 seems to be marginal, thus more justification of the relevance of this research is suggested. I recommend to extend the Introduction by more up-to-date references to show why this work may be important for the readers of Nanomaterials. In particular, what specific application do the authors see for CaSi2 crystals either directly grown on Si or formed by irradiation of a CaF2 layer?
Response 1: We expanded the introduction and added links to modern works on this topic (page 1). «Investigations the growth and properties of calcium silicides are currently being carried by some groups of reserchers and calcium silicide films were obtained on Si(111) and Si(001) surfaces with various phase compositions. It has been found that when Ca and Si are co-deposited on a hot substrate, with increasing of substrate temperature, there is a tendency to form a phase with the highest silicon content: Ca2Si-CaSi-CaSi2.[1-3] It has been established that Ca2Si films are a narrow-gap semiconductor material, while CaSi and CaSi2 films are semimetals. Ca2Si was shown the best thermoelectric properties [4-7]. Approaches are being developed for obtaining materials with different functional properties: translucent and conductive CaSi2 films for silicon electronics and optoelectronics [7–10]. Recently we report on the studies of the 20 keV electron-beam irradiation(current density 50 A/m2) effects on the epitaxial CaF2 film growth on a Si surface.It was found that the area exposed to the electron beam during epitaxially growingCaF2 film CaSi2 layer synthesis Radiation-stimulated CaSi2 growth allow preparing local CaSi2 films on a Si wafer, the sizes of CaSi2 regions being limited by just by size of the focused electron beam only and, also, in the possibility to control the CaSi2 film thickness with accuracy of one monolayer by varying both the radiation dose and the initial Si layer thickness and the sacrificial CaF2 layer[8].»
Point 2: Please provide more details about the MBE apparatus, RHEED, electron gun or at least indicate a reference where these details are available. What was the spot size of the electron gun? Was the spot scanned over the surface? From the text it may appear that the RHEED electron source was used for irradiation; please confirm.
Response 2: We have used home made developed and produced of system MBE "Katun-100" details of which can be found on our website [https://www.isp.nsc.ru/16/Renew/pgs/Laboratory/K-100.html]. We used for electon irradiation RHEED electron source of MBE equipment. The electron beam spot sise was 2 mm, the angle betweem direction of electon beam and substrate plane was 40. There was no spot scannned over serface.
Point 3: Results, Figs 3-4: even if the qualitative conclusion that irradiation during deposition results in directed crystal growth while post-deposition irradiation leads to formation of randomly oriented crystallites is supported by the data, the presentation is very confusing. Especially, the meaning of the X,Y, X' and Y' axes, their relation to laboratory and Si substrate-fixed coordinate systems is unclear. The meaning of the "2" or "3" numbers in the peak assignments is also unclear. If the Eg mode is at 413 cm-1, why peaks appear at 418 cm-1? I would recommend to reformulate this part (maybe along with a more mathematically correct description of the geometry- and polarization dependence of the Raman intensities), and if it is too complicated or lengthy, to present it as supplementary information.
Response 3:Thank You very much for relevant remark, we hope that our answer makes the text more clear. This paragraph has been added to the text of the article (page 5).
«In polarization measurements of Raman scattering, the so-called Porto notation is used. In our case, backscattering geometry was used, the wave vector of the incident light was directed along the Z axis (i.e., along the (111) crystallographic direction in the Si substrate). The wave vector of the scattered light was directed along the -Z axis. The light wave is transverse, so the directions of polarization of the incident and scattered light are perpendicular to the Z axis. In Porto XY notation, the first direction is the polarization of the incident wave, and the second direction is the polarization of the scattered wave. In our case, the X axis corresponds to the crystallographic direction (1-10), the Y axis corresponds to the crystallographic direction (11-2). One can see, that X, Y and Z axes are orthogonal. In result of 45o rotation, the X’ axe are bisector of the angle X0Y, the Y’ axe are bisector of the angle -X0Y. The XY (or X’Y’) polarization geometry is called transverse (the polarization of scattered light is perpendicular to the polarization of the incident light), and the YY (or Y’Y’) polarization geometry is called longitudinal, the polarization of light does not change upon scattering.»
As for difference between observed frequency of Eg mode (418 cm-1) and the literature data (413 cm-1), one can assume, that it can be due to strain. It is known, that due to anharmonism, phonon frequencies depend on strain. Usually, the compressive strain leads to blueshift in phonon frequencies. This paragraph has been added to the text of the article (page 6).
«As for difference between observed frequency of Eg mode (418 cm-1) and the literature data (413 cm-1), one can assume, that it can be due to strain. It is known, that due to anharmonism, phonon frequencies depend on strain. Usually, the compressive strain leads to blueshift in phonon frequencies.»
Figs 2,3,4. The numbers "2" and "3" were the numbers of the spectra, they were removed because they do not carry information, but only complicate the understanding of the figure, thanks for the comment.
Point 4: Figures 6,7: maybe because of the lower quality images provided for review, I'm unable to see the semi-circles. Moreover, the red circles in Fig. 6b make it really difficult to compare the two patterns and to recognize the splitting of the reflections. Please try to resolve the situation!
Response 4: Figures 6 have been corrected. We tried our best to improve the quality of the drawings. Since in this work we studied thin films (10 nm), the features in the RHEED patterns appear weakly (semicircles, double reflections). To see the semi-circles, we had to increase the brightness in the photo to the maximum, but double reflections cannot be seen. Therefore, a photo with increased brightness is shown as an inset to Figure 6
Point 5: Although the material formed during electron irradiation of CaF2 was investigated by several different techniques, phase identification is based solely on the Raman spectra. How formation of mixed compounds (like CaSi(2-x)Fx) or silicides with different stoichiometry is ruled out? What happens with the material upon air exposure (as far as I understand, Raman measurements were performed ex situ).
Response 5: Previously, we studied [Aleksey V. Kacyuba, Anatoly V. Dvurechenskii, Genadiy N. Kamaev, Vladimir A. Volodin, Aleksey Y. Krupin. Crystal structure of thin CaSi2 films grown by radiation-induced epitaxy. Journal of Crystal Growth 2021, 562 , 126080. https://doi.org/10.1016/j.jcrysgro.2021.126080] crystal structure of thin CaSi2 films grown by radiation-induced epitaxy. The dependence of various CaSi2 polymorphs (3R and 6R) on the thickness of the CaF2 film is shown. For thin films crystal structure of CaSi2 films belongs to the trigonal rhombohedral modification 3R (space group R3‾m with a three-layer translational period of silicon substructures in the unit cell 3R). The transition of CaSi2 crystal structure to the trigonal rhombohedral modification a six-layer period material 6R was found for thicker films (>20 nm). In this work, we study the formation of CaSi2 films during the action of an electron beam on CaF2 both directly during the epitaxial growth of CaF2 and after the formation of a CaF2 film 10 nm thick. Therefore, we exclude the formation of silicides of different stoichiometry.
Formation of mixed compounds (such as CaSi(2-x)Fx) or silicides of different stoichiometry, depend on the temperature of the sample[1-3]. In our experiments, the temperature is obviously higher than that when mixed compounds are formed.
Point 6: For me a Discussion section is missing. This section would offer opportunity to explain the results, analyze the effect of the different conditions on the film structure and to relate the observations to previous investigations. For example, a relevant question would be the effect of temperature or the electron energy on the silicide formation upon electron beam irradiation. The text currently in Lines 252-261 belongs to this Discussion section. As a side note, do the results presented here imply that RHEED cannot be used for analysis of the growth of CaF2 on Si because of the radiation-induced silicide formation?
Response 6: Discussion section is added. "It was found that irradiation during the epitaxial CaF2 film growth leads to the heterogeneous nucleation and growth of Si-related nanowhiskers CaSi2, oriented along the direction <1-10> with average size 5 um.
Irradiation of the formed epitaxial CaF2 film with a thickness of 10 nm at a similar electron irradiation dose and substrate temperature leads to the random CaSi2 nucleation centers in CaF2 film with a subsequent crystallite proliferation. Analysis of the RHEED and SEM patterns shows that disordered CaSi2 crystallites are present on the surface. This indicates the homogeneous nucleation of CaF2. Various methods of the CaSi2 (L1, L2) synthesis for thin films (10 nm) lead to the formation of the CaSi2 polymorph 3R, despite the fact that this phase is metastable, compared to the 6R polymorph. The reason of growth the CaSi2 polymorph 3R, metastable phase apparently is that the Electron beam irradiation is strongly no equilibrium асtion as during irradiation and further relaxation with rather short time to form more complicate structure of 6R-CaSi2 (stable phase). For thick CaF2 films deposition( more 20 nm) electron irradiation lead to the formation of the CaSi2 as a stable 6R polymorph [8]."
The first experimental data on the formation of CaSi2 films during the action of an electron beam on CaF2 after the formation of a CaF2 film with a thickness of more than 20 nm have already been obtained and analyzed. We hope to share our results in new publications in the near future.
As for your question regarding the possibility of using RHEED for analysis of the growth of CaF2 we think that RHEED cannot be used for analysis of the growth of CaF2 on Si. Moreover, there are data indicating that under the influence of an electron beam, the phenomenon of radiolysis is observed - the dissociation of films into calcium and fluorine.[A. A. Velichko, V. A. Ilyushin, D. I. Ostertak, Yu. G. Peisakhovich ,N. I. Filimonova. Effect of high-energy electron beam irradiation on the surface morphology of CaF2/Si(100) heterostructures, Journal of Surface Investigation. X-ray, Synchrotron and Neutron Techniques volume 1, pages479–486 (2007)]
Point 7:Lines 36 and 69: the irradiation current density was 50 uA/m2 or 50 uA/cm2?
Response 7: The current density value is corrected to 50 µA/m2. Thank you for your comment .
Point 8: Lines 80, 86 (Figure 1): with what technique was the image taken? If I understand correctly, the lines represent the irradiated areas, while L0 is the dark uniform area around and between the lines. If it is correct, the white-green dot is confusing.
Response 8: Figure 1 has been corrected. Indeed, the lines in Fig.1 are irradiated areas. The photograph of the sample was taken on a canon m6 camera with a standard 15-45mm lens. The insert to Fig. 1 was made on an optical microscope "Optem zoom 2.0"
Point 9:Figure 2: the notation for the (missing) A1g(Ca) peak is misleading; maybe an arrow to its expected position would be helpful.
Response 9:Figure 2 corrected. Thank you for your comment.
Point 10: Figure 5: was the diffraction pattern recorded after oxide removal or buffer layer growth?
Response 10:The RHEED pattern was recorded after the removal of the protective oxide and the growth of the Si buffer layer.
«Figure 5.RHEED image of the Si substrate surface after removing a protective oxide and the Si buffer layer growth.»
Point 11: Line 230: please resolve the abbreviation NW.
Response 11: Nanowiskers (sorry for unclear writing, we will check paper. )
Point 12: In general, the English usage throughout the paper is appropriate. However, some revision is still needed. For example:
Line 76: the word "structures" refers to what? Line 80: "image of the studied samples" is the correct wording. Line 90: "recorded outside of the" would be correct. Line 101: "the line at 305 cm-1" would be correct. Lines 102, 111: repeating the same information is unnecessary. Line 158, 167, etc.: please write "reflections" instead of "reflexes". Line 242: "a" is not needed. Lines 257-259: please reconsider the sentence.
Response 12: We have updated the article. Thank you for remarks.

Reviewer 2 Report
The article “Radiation-induced nucleation and growth of CaSi2 crystals both directly during the epitaxial CaF2 growth and after the CaF2 film formation”, by A. V. Dvurechenskii et al., studies the effects of irradiation with fast electrons on the growth of CaSi2 crystals, and its interaction with the epitaxial CaF2 film precursor. The article is well written, with a clear presentation of the growth and characterization methods, along with a clear interpretation of the experimental results. In particular, the morphological features presented in the SEM images of Fig.10, provide a convincing argument to distinguish between the effects of irradiation during the CaF2 film growth, as compared to the irradiation posterior to the film formation. This observation is further supported with AFM images, as presented in Fig. 11 and Fig. 12. Based on this evidence, the authors conclude that irradiation during the film growth leads to Si-related nanowhiskers of CaSi2, while irradiation of the formed epitaxial CaF2 film leads to a more chaotic crystallite formation.
In summary, the study is relevant for the field of CaSi2 crystal growth, and provides a well suported comparison between two alternative electron irradiation procedures that lead to clearly different final crystalline morphologies. I recommend the paper for publication in its present form in the journal Nanomaterials.
Author Response
Thank you for clear comments
Reviewer 3 Report
The manuscript can be accepted with minor revison.
Author Response
We have updated the article. Thank you for clear comments
Round 2
Reviewer 1 Report
In my opinion the authors adequately addressed my issues in their Response. On the other hand, the responses were not always incorporated into the text of the manuscript. Thus I recommend the acceptance of this work for publication if the missing information listed below appears also in the text.
Line 172: although the details of the MBE apparatus are identified in the response, the reference is still missing from the manuscript; please add it. Please also indicate in lines 181-182 that the RHEED source was used for irradiation.
Lines 217, 224: please indicate that the image is an optical micrograph
Line 538: I would recommend to extend the discussion by indicating the problem of RHEED study of CaF2 epitaxial films on Si as it is described at the end of Response No. 6 to my previous comments.
Minor improvements still needed:
Line 67: please write "Investigations of the growth...". Line 77: please check the current density. Lines 77-79: the sentence seems to be incomplete. Line 79: "allows" instead of "allow". Line 81: "has the possibility" seems to be better. Lines 158-159: "polymorph structures with a three-layer..." seems to be better. Line 441, 443 and below: please write "reflection"instead of "reflex". Line 592: "growth of metastable..." seems to be better. Line 593: "non-equilibrium" seems to be better.
Author Response
Point 1: Line 172: although the details of the MBE apparatus are identified in the response, the reference is still missing from the manuscript; please add it. Please also indicate in lines 181-182 the RHEED source was used for irradiation.
Response 1: Reference added to the manuscript. In lines 181-182 the RHEED source was added. Thank you very much for relevant remark.
Point 2:Lines 217, 224: please indicate that the image is an optical micrograph
Response 2: thanks corrected
Point 3:Line 538: I would recommend to extend the discussion by indicating the problem of RHEED study of CaF2 epitaxial films on Si as it is described at the end of Response No. 6 to my previous comments.
Response 3: This paragraph has been added to the to the manuscript to the discussion section
It must be noted, that RHEED problematic to use for analysis of the growth of CaF2 on Si because of area exposed to the electron beam suffers strong modifications, such as in the surface morphology and film chemical composition. There are information indicating that under the influence of an electron beam, the phenomenon of radiolysis is observed - the dissociation of films into calcium and fluorine[26].
Point 4: Line 67: please write "Investigations of the growth...".
Response 4: thanks corrected
Point 5:Line 77: please check the current density.
Response 5: thanks corrected
Point 6:Lines 77-79: the sentence seems to be incomplete.
Response 6: thanks corrected. The sentence is changed
It was found that the area exposed to the electron beam during the epitaxial growth of the CaF2 film is the synthesized CaSi2 layer.
Point 7:Line 79: "allows" instead of "allow".
Response 7: thanks corrected
Point 8:Line 81: "has the possibility" seems to be better.
Response 8: thanks corrected
Point 9:Lines 158-159: "polymorph structures with a three-layer..." seems to be better.
Response 9: thanks corrected
Point 10:Line 441, 443 and below: please write "reflection"instead of "reflex".
Response 10: changes made throughout the manuscript
Point 11:Line 592: "growth of metastable..." seems to be better.
Response 11: thanks corrected
Point 12:Line 593: "non-equilibriu
Response 12: thanks corrected
Thank you very much for relevant remark.
